# Effects of Storage Conditions on Degradation of Chlorophyll and Emulsifying Capacity of Thylakoid Powders Produced by Different Drying Methods [note 1]

**DOI:** 10.3390/foods9050669

**Published:** 2020-05-22

**Authors:** Karolina Östbring, Ingegerd Sjöholm, Marilyn Rayner, Charlotte Erlanson-Albertsson

**Affiliations:** 1Department of Food Technology, Engineering and Nutrition, Lund University, 221 00 Lund, Sweden; ingegerd.sjoholm@food.lth.se (I.S.); marilyn.rayner@food.lth.se (M.R.); 2Department of Experimental Medical Science, Appetite Control Unit, Lund University, 221 00 Lund, Sweden; charlotte.erlanson-albertsson@med.lu.se

**Keywords:** thylakoids, functional foods, drying, chlorophyll degradation, emulsifying capacity, processing

## Abstract

Thylakoid membranes isolated from spinach have previously been shown to inhibit lipase/co-lipase and prolong satiety in vivo. There is a need to develop thylakoid products that not only have the desired characteristics and functionality after processing, but also are stable and provide equivalent effect on appetite over the promised shelf life. The aim of the present study was therefore to evaluate how the thylakoid powders’ characteristics and functionality were affected by moisture during storage. Thylakoids produced by drum-drying, spray-drying, and freeze-drying were incubated in controlled atmosphere with different relative humidity (10 RH%, 32 RH%, 48 RH% and 61 RH%) for 8 months. The water content in all powders was increased during storage. The water absorption was moisture-dependent, and the powders were considered hygroscopic. Relative humidity showed a definite influence on the rate of chlorophyll degradation and loss of green color in thylakoid powders after storage which correlated with impaired emulsifying capacity. Spray-dried powder had the overall highest chlorophyll content and emulsifying capacity at all RH-levels investigated. Spray drying was therefore considered the most suitable drying method yielding a powder with best-maintained functionality after storage. The results can be applied towards quality control of high-quality functional foods with appetite suppressing abilities.

## 1. Introduction

Dehydrated foods are subjected to physical, chemical and biological deterioration processes during storage such as loss of nutritional substances, color, and aroma compounds [1]. Water activity is a key parameter that determines storage stability of dry products. Lipid oxidation is accelerated in materials with very low water activity [2] and enzymatic reactions and growth of undesirable microorganisms are accelerated at high water activity during storage [3]. For functional foods containing active ingredients, it is important to control these processes during storage to ensure both microbial safety and the proposed function over the products’ shelf life.

Thylakoid membranes isolated from spinach can be used as an appetite-reducing ingredient promoting weight loss due to increased satiety between meals [4]. Thylakoids are the photosynthetic membrane in the chloroplast and is responsible for the conversion of solar energy into adenosine triphosphate (ATP) and nicotinamide adenine dinucleotide phosphate (NADPH), used in the production of carbohydrates [5]. Membrane lipids such as galactolipids, phospholipids and sulpholipids account for approximately 30% of the thylakoids mass. The remaining 70% is membrane proteins together with their bound pigments i.e., chlorophyll, carotenoids and xanthophylls, involved in the photosynthetic electron transport. The membrane proteins are both extrinsic, i.e., located on the membrane surface and intrinsic, i.e., membrane spanning [6].

Thylakoid membranes expose hydrophilic, hydrophobic and charged groups [7] and the complex structure is suitable for lipid containing food formulations such as emulsions. The membrane protein complexes attach to the oil-water interface with high binding capacity [7,8]. The specific component providing the interfacial properties is suggested to be a part of the intrinsic proteins, namely the hydrophobic alpha helices found in the light harvesting complexes [9]. Thylakoids can thereby adsorb to the lipid surface by hydrophobic interactions, creating a barrier around the lipid droplets [9]. The barrier thereby hinders gastrointestinal lipase/co-lipase from coming into close proximity with their substrate, reducing the rate of lipolysis.

Prolonged lipolysis has been demonstrated to suppress appetite through the ileal brake mechanism. Accumulation of undigested lipids and lipolysis products in ileum promotes secretion of satiety promoting hormones that slows down gastric emptying, reduce appetite and in turn decrease food intake [10]. Thylakoids have been demonstrated to prolong lipolysis in vitro under duodenal conditions [11] as well as in vivo in animal models [12,13]. Supplementation of thylakoids to the diet in acute meal studies elevated the satiety hormones cholecystokinin (CCK), glucagon-like peptide 1 (GLP-1), leptin and enterostatin while the hunger hormone ghrelin was reduced [14,15]. Thylakoids have also been demonstrated to induce weight loss and urge for palatable food in humans in a three months study [4].

The powder formula offers several advantages over aqueous solutions in terms of flexibility in food applications as well as reduced transport and handling costs [16]. Further, the shelf life can be significantly prolonged by reduction of water content [17]. To find the optimal drying technique in terms of maintenance of the thylakoid’s functionality, an evaluation of three different drying techniques was performed. In a previous study, thylakoids were drum-dried, spray-dried and freeze-dried and the effect on powder characteristics and functionality was evaluated [18]. Freeze-dried thylakoids (25 °C for 7 days) had the highest chlorophyll content, the highest emulsifying capacity and the highest ability to inhibit lipase/co-lipase followed by spray-dried (inlet temperature 120 °C/outlet temperature 72 °C with residence time a few seconds) and drum-dried thylakoids (105 °C for several seconds). The loss of emulsifying capacity was correlated to degree of heat treatment during drying. It was suggested that heat treatment induced chlorophyll degradation and chlorophyll, in turn, is known to support the structure of the light-harvesting complex housing the alpha helices. Chlorophyll degradation facilitated aggregation of the thylakoids, with reduced interfacial properties hence reduced ability to inhibit lipase/co-lipase as consequence.

There is a need to develop thylakoid products in form of powders that not only have the desired functionality after drying, but also are stable and provide the same effect on appetite over the promised shelf life. The thylakoids’ functionality is linked to the chlorophyll content thus it is important to maintain the chlorophyll content in the powder during processing and storage to as great extent as possible. Chlorophyll is sensitive to heat, acid and enzymatic degradation by chlorophyllase [19,20]. Furthermore, chlorophyllase activity is known to be accelerated by high relative humidity [21]. If a powder shall be stored with maintained bio functionality, it is important to determine how moisture from the surrounding environment affects the active ingredient. When these parameters are known, it is possible to design a package protecting the thylakoids’ function during storage. The aim of the study was therefore to evaluate to what extent exposure to moisture during storage affect the thylakoids’ characteristics and emulsifying capacity. The present work investigated the influence of different relative humidity on thylakoid powders dried by spray drying and freeze drying and was compared with two commercial thylakoid products. The results can be applied in the quality assurance of dehydrated functional food products.

## 2. Materials and Methods

### 2.1. Isolation of Thylakoid Membranes and Powder Production

Thylakoids were isolated as previously described by Östbring et al. [18]. Frozen spinach leaves (*Spinacia oleracea*) were homogenized with water in a blender (Robot Coupe Blixer 4 Robot Coupe SA, Bourgogne, France). The slurry was further homogenized at 60 Hz for three minutes (Ytron-Y Jet Stream, Ytron, Bad Endorf, Germany) and filtered through four layers of 20 µm polyester mesh filter. The filtrate was centrifuged at 5000× *g*, 4 °C, for 30 min (Beckman Coulter Allegra X-15 R Centrifuge, Fullerton, CA, USA). The sediment was collected and re-suspended in water (0.011 kg dry solids/kg water), and was thereafter homogenized (Ystral D-79282, Ballrechten-Dottingen, Germany) until a homogenous slurry was obtained. The pH was adjusted to 7.0 with NaOH.

Thylakoid slurry was either spray-dried or freeze-dried. Spray drying was performed using a lab scale Büchi spray drier (B-290, Büch Labortechnik AG, Flawil, Switzerland). Drying conditions during spray drying were inlet air temperature was 120 °C, outlet temperature 72–75 °C and feed flow rate 0.6 L/h. Freeze drying was carried out using a laboratory freeze dryer (Hetosicc freeze dryer CD 12, Birkerod, Denmark). The sample were frozen in −18 °C for 24 h prior to drying. The plate temperature was 20 °C, the condenser −50 °C and the pressure in the freeze dryer was 0.02 mbar. Residence time in the freeze-dryer was seven days. The freeze-dried powder was grinded in an electrical coffee grinder (OBH Nordica) to reduce the particle size. The resulting powders were stored dark and in sealed containers in −18 °C until storage incubated in controlled atmosphere. Spray-dried powders are hereafter referred to as SD_w_ and freeze-dried powders as FD_w_.

### 2.2. Commercial Thylakoid Samples

Two types of commercial thylakoid samples were received from Green Leaf Medicals and the isolation process was essentially as described by Emek et al. 2009 [6]. Hot air-dried spinach leaves (*Spinacia oleracea*) were homogenized with water (1:1 *w*/*w*) followed by filtration. The filtrate was diluted with water (1:9 *w*/*w*) and the pH was adjusted to the thylakoid membranes isoelectric point 4.7. The filtrate was incubated at 4 °C for four hours. The sediment was collected and once again diluted with water (1:9 *w*/*w*), pH was adjusted to 4.7 and the dispersion was incubated as described above. The sediment was collected, and pH was adjusted to 7.0 with NaOH. Thylakoids were thereafter drum dried under industrial conditions. Typical drum drying conditions have a drum surface temperature of 105 °C with drying completed within 30 s. One of the received samples was produced with maltodextrin (1:3 *w*/*w* dry base) and are hereafter referred to as drum-dried thylakoids produced by the pH method with addition of maltodextrin (DD_pH+M_). The other sample was produced under similar conditions but without the addition of maltodextrin and are hereafter referred to as drum-dried thylakoids produced by the pH method (DD_pH_).

### 2.3. Incubation in Controlled Atmosphere

Saturated salt solutions of LiCl, MgCl_2_, Mg(NO_3_)_2_ and KI were prepared and put in desiccators to achieve a relative humidity of 10 RH%, 32 RH%, 48 RH% and 61 RH% respectively. The desiccators were left for at least one week to reach equilibrium. All powders (DD_pH_, DD_pH+M_, SD_w_, FD_w_) were freeze-dried prior to incubation in the desiccators to assure a low initial water activity (see above for details). After freeze-drying, three grams of each powder were put in open Petri dishes in the different desiccators, four dishes in each desiccator in total. The powders were incubated dark in 20.0 °C for a total of eight months.

### 2.4. Evaluation of Powder Characteristics

A series of analysis was performed to evaluate the effect of thylakoid powder during storage in controlled atmosphere. Analysis was performed both prior to incubation and after incubation for eight months in desiccators with specific humidity.

#### 2.4.1. Water Activity

Water activity was analyzed using a water activity meter (AquaLab Ver 3TE, Decagon Devises, Pullman, WA, USA). The equipment was calibrated with standard salt solutions (13.41 (0.25 a_w_) or 8.57 (0.50 a_w_) mol/kg LiCl). Analysis was performed in triplicates.

#### 2.4.2. Dry Matter Analysis

Dry matter content was determined according to the official method of analysis (AOAC) by drying in a convective oven at 102 °C for 24 h or longer until a constant weight was obtained. Analysis was performed in duplicates.

#### 2.4.3. Color Analysis

The color attributes (Hunter L, a and b values) of each powder were measured with a portable spectrophotometer (CM-700d, Konica Minolta, Osaka, Japan). Each sample was randomly measured at three locations and the average was reported.

#### 2.4.4. Chlorophyll Content

Chlorophyll content in the thylakoid powders were determined by photo spectroscopy according to Porra et al. [22]. Powders (0.04–0.08 g) were homogenized with 2 mL deionized water in a Potter-Elvehjelm homogenizer until a homogenous slurry was obtained. 10–100 μL thylakoid slurry was added to 1 mL ice-cold acetone (80 vol%) depending on color intensity of the powder. The samples were vortexed and incubated dark and on ice for 20 min followed by centrifugation (Eppendorf Mini Spin, Eppendorf AG, Hamburg, Germany) at 13,400× *g* for 4 min, at 25 °C before spectrophotometric analysis was carried out. Spectra were obtained over the wavelength range 200–800 nm against a blank of acetone (80 vol%). Absorbance at λ = 646.6 nm and λ = 663.6 nm was used to quantify the chlorophyll content. Chlorophyll a and total chlorophyll concentration were analyzed in triplicate and determined according to Porra et al. [22]:Chlorophyll a = 12.25 A_663.6 nm_ − 2.550 A_646.6 nm_
Chlorophyll a + b = 17.76 A_646.6 nm_ + 7.340 A_663.6 nm_

### 2.5. Evaluation of Powder Functionality

Thylakoids are known to adsorb to the oil-water interface due to exposure of hydrophobic, hydrophilic and charged side groups. In previous studies close correlations were found between thylakoids’ interfacial properties, lipase-inhibiting capacity and chlorophyll content (*R*^2^ = 0.80 for interfacial properties vs. lipase-inhibiting capacity, *R*^2^ = 0.81 for interfacial properties vs. chlorophyll content, *R*^2^ = 0.95 for chlorophyll content vs. lipase/inhibiting capacity) [23,24]. Due to the close correlation, the thylakoids ability to stabilize the oil-water interface of oil-in-water emulsions was used as a functional endpoint in the present study. To examine whether exposure to varying humidity affected the thylakoid powders interfacial properties, the thylakoids’ ability to stabilize the oil-water interface were analyzed both prior to incubation in controlled atmosphere and after 8 months storage.

#### 2.5.1. Preparation of Thylakoid-Stabilized Emulsions

Emulsions were prepared in glass test tubes by adding 64 mg thylakoid powders (dry basis) dried by either drum drying, spray drying or freeze drying to 2 mL continuous phase (5 mM phosphate buffer with 0.2 M NaCl, pH 7.0) and 1 mL lipid phase (Miglyol 812, Asaaol AG, Germany). The mixture was homogenized by a high shear mixer with a 6 mm dispersing head (Ystral D-79282, Ballrechten-Dottingen, Germany) operated at 22,000 rpm for 60 s. The emulsions were prepared in triplicates and were incubated dark in 4 °C for 60 min prior to particle size distribution analysis.

#### 2.5.2. Particle Size Measurements of Thylakoid-Stabilized Emulsions

Particle size distribution of the emulsions stabilized by thylakoid powder were analyzed with a laser diffraction particle analyzer (Malvern Mastersizer 2000 Ver 5.60, Malvern, Worcestershire, UK). The dispersing unit (Hydro 2000S) was filled with 100 mL MilliQ water and the pump was operated at 2000 rpm. The glass test tubes were turned upside down three times before a small volume was added to the flow system and pumped through the optical chamber for measurements. Obscuration was between 10% and 20%. The RI of the sample was set to 1.45 (miglyol) and the RI of the continuous phase was set to 1.33 (water). The emulsifying capacity (EC) of the thylakoid membranes i.e., the maximum surface that can be created and stabilized by a unit emulsifier (m^2^/mg) was calculated as
EC = 6ϕ/(C_a_∙*d*_32_)
where ϕ is the disperse phase volume fraction, C_a_ is the initial concentration of emulsifying agent (assumed that all emulsifier is adsorbed at the oil-water interface) and *d*_32_ is the volume-surface mean droplet diameter.

### 2.6. Statistical Analysis

All statistical analyses were performed using SPSS software version 22 (IBM). For parametric datasets (L-, a-, and b values, total chlorophyll and emulsifying capacity), the univariate general linear model with Tukey’s test was performed to investigate significant differences. For non-parametric datasets (dry matter, water content and water activity), a Kruskal–Wallis test with Bonferroni-adjusted pairwise comparison was performed. Differences were considered significant if *p* < 0.05

## 3. Results and Discussion

The present work investigates the effect of storage conditions in varying relative humidity on appetite reducing thylakoid powders produced and dried in different ways. Powder characteristics and functionality, i.e., dry matter, water activity, chlorophyll content, color and emulsifying capacity were evaluated after 8 months storage.

### 3.1. Water Content and Water Activity

The water content for the majority of the powders (DD_pH+M_, SD_w_ and FD_w_) increased during 8 months of storage in controlled atmospheres with different relative humidity. The commercial sample DD_pH_ had a higher water content prior incubation compared with the other powders and was instead subjected to desorption when incubated in 11 RH%, 32 RH% and 48 RH% but after incubation in the highest relative humidity (61 RH%), absorption was taking place in these samples as well. Increase in water content during storage can either depend on absorption or chemical or biological reactions in the material. In the present study, the weight of the samples increased over time indicating that water absorption was the dominating reaction.

During the storage period of 8 months, equilibrium was reached for the drum-dried powder with addition of maltodextrin (1:3 *w*/*w* dry basis) but not for the other powders. Drum-dried thylakoid powders (DD_pH_) displayed a sigmoidal shape with the highest increase in water content between 32 RH% and 48 RH%. The water content varied from 0.0494 kg water/kg solids at 10 RH% to 0.1017 kg water/kg solids at 61 RH% and the change was significant. DD_pH+M_ had lower water content compared to the other powders with 0.0322 kg water/kg solids at RH 10% to 0.1025 kg water kg water/kg solids at 61 RH% although not significantly compared with other samples incubated under the same conditions, and the sorption isotherm followed the characteristic behavior of maltodextrin. Spray-dried and freeze-dried powders had higher water content at higher relative humidity levels, compared to the drum-dried powders, although not significant. Water content varied from 0.0528 kg water/kg solids (10 RH%) to 0.1173 kg water/kg solids (61 RH%) for spray-dried powders and 0.0548 kg water/kg solids (10 RH%) to 0.1291 kg water/kg solids (61 RH%) for freeze-dried powders.

All powders, independent on drying method displayed higher water activity when exposed for higher relative humidity (Figure 1). Spray-dried and freeze-dried powders appeared to display water activities in a narrower range compared to drum-dried powders. There was a trend towards higher water activities for spray-dried and freeze-dried powders (a_w_ = 0.30 and 0.29) compared to drum-dried powders (a_w_ = 0.24 for both) after exposure for the lowest relative humidity (RH 10%) investigated, but the difference was not significant. At the highest relative humidity (RH 61%) the trend was reversed with spray-dried and freeze-dried powders displaying lower water activities (a_w_ = 0.38 and 0.46) compared to drum-dried powders (a_w_ = 0.51 for DD_pH_ and 0.58 for DD_pH+M_), although not significant. The differences may be explained by different morphology and porosity of the powders, although the mentioned parameters were outside the scope of the present study.

### 3.2. Color Loss During Storage Are Related to Specific Relative Humidity

Storage for 8 months affected the lightness and green color of the powders (Table 1, Figure 2). The L-value was significantly decreased (indicating darker color) for DD_pH_ and SD_w_ when the lightness prior incubation was compared with lightness after incubation in 48 RH% (SD_w_) and 61 RH% (both DD_pH_ and SD_w_). When the lightness between powders in similar relative humidity was compared, FD_w_ had in general lower L-values compared to the other powders in all relative humidity’s investigated and additionally SD_w_ was significantly darker compared to DD_pH_ and DD_pH+M_ in the lowest relative humidity’s.

All powders displayed an increased a-value (indicating loss of green color) after incubation and the changes were dependent on the specific relative humidity, with a greater loss of green color when exposed for higher relative humidity. For DD_pH_ and DD_pH+M_ the loss of green colour was only significant for samples incubated in 61 RH% whereas for SD_w_ both incubation in 48 RH% and 61 RH% resulted in a significant loss of green color. For FD_w_, incubation in all relative humidity’s resulted in significant differences where incubation in higher relative humidity’s was associated to greater loss. The spray-dried and freeze-dried powders had initial lower a-values (indicating more intense green color) compared to drum-dried powders and were consequently displaying a larger reduction in green color at all relative humidity levels investigated. The effect was more pronounced for powders exposed to higher relative humidity. The b-value was changed after incubation and the powders significantly lost yellow color after incubation in 61 RH% except for DD_pH+M_ where there was no difference.

### 3.3. Chlorophyllase Catalyze Degradation of Chlorophyll

In dehydrated foods, deteriorative reactions depend on the state of water present [21]. At very low water activities (a_w_ = 0–0.2), a monolayer of water is tightly bound to ionic groups of the food material such as anion and carboxyl groups [25]. At slightly higher water activities (a_w_ = 0.2–0.4) the water exists as multimolecular layers, the fraction being less firmly bound compared to the monolayer at very low water activities. At high water activity (a_w_ = 0.4 and above), water exists as free molecules resulting in accelerated undesirable enzyme reactions and microbial growth [3].

Chlorophyllase, the enzyme catalyzing the conversion of chlorophyll to its degradation product chlorophillide has been reported to be accelerated by high water activity [21]. Chlorophyllase activity has also been reported to be temperature dependent with a maximum at 23 °C [26]. Many efforts have been invested in controlling chlorophyllase to prevent discoloration and prolong the shelf life of green vegetables. Chlorophyllase is known to be accelerated by mild heat treatment (50 °C), however the activity has been reported to be seriously reduced upon heating above 80 °C [27]. Ihl et al. reported that blanching for a few minutes in boiling water inactivated the chlorophyllase in artichokes [20].

Relative humidity showed a definite influence on the rate of chlorophyll degradation in thylakoid powders stored for 8 months (Figure 3). Chlorophyll content was dramatically degraded in spray-dried and freeze-dried powders after storage with a greater chlorophyll loss at exposure for higher relative humidity. For spray-dried powders, there were no significant change at low relative humidity (10 RH% and 32 RH%), but at higher relative humidity, the chlorophyll content significantly decreased from 68 mg/g prior incubation to 42 mg/g (−38%) and 32 mg/g (−52%) after incubation in 49 RH% and 61 RH%. Freeze-dried powders displayed the overall largest chlorophyll degradation of the powders investigated. The chlorophyll content decreased from 76 mg/g prior incubation to 58 mg/g and 54 mg/g (11 RH% and 33 RH%, no significant difference between the two) and when powders was incubated in 49 RH% and 61 RH%, the concentrations was even lower: 39 mg/g (−48%) and 27 mg/g (−63%) after incubation in 49 RH% and 61 RH%.

The drum-dried powder did not have a significant amount of chlorophyll prior to incubation, hence no significant chlorophyll degradation.

The spray-dried and freeze-dried powders displayed absorption spectra with a shift of maximum in the blue spectrum to shorter wavelengths after storage. The height of the peak corresponding to chlorophyll *a* (at 436 nm [19]) was reduced with increased relative humidity, and the peak corresponding to the degradation products pheophorbide *a* and pheophytin *a* (both displaying absorption maximum at 409–415 nm [19]) was present independent of relative humidity during storage (Figure 4). The height of the peak corresponding to chlorophyll *a* was decreased to some extent for the drum dried powders (Figure 4). These findings are in accordance with Dutton et al., where chlorophyll degradation in dehydrated spinach was reported to be dependent on relative humidity during storage [28]. Dutton et al. reported a shifted absorbance maxima towards the blue region and decreased maximum at 665 nm after storage in high relative humidity [28]. The changes were attributed to chlorophyll degradation. Chlorophyll can be degraded via two different routes: chemical degradation or enzymatic degradation. Chlorophyll is degraded to pheophytin by heat or acid by replacement of the central Mg^2+^ ion by two H^+^, which induce a color shift from bright green to olive brown [29]. The mechanism is not clear but magnesium dechelatase are proposed to be involved in the ion replacement [30]. The enzymatic degradation route involves chlorophyllase catalyzed removal of the chlorophyll’s phytol chain producing chlorophillide with a bright green color displaying the same absorption maximum as chlorophyll (436 nm). Chlorophillide is rapidly degraded by heat or acid to the olive brown pheophorbide by replacement of the central Mg^2+^ ion by two H^+^ [31]. The two degradation products pheophytin and pheophorbide cannot be separated by photo spectrometric analysis [32]. Degradation products (pheophorbide and/or pheophytin) were detected in the present study (Figure 4). Heat or acid is required for formation of these products. Dutton et al. reported the pH in dehydrated spinach to be lowered during storage in high humidity [28]. Since no heat was applied in the system in the present study, the degradation is suggested to be driven by protons probably provided from dissociated molecules like carotenoids in the system.

### 3.4. Heat Treatment During Drying Affects Chlorophyll Loss During Storage

The thylakoid powders investigated in the present study had been subjected to different heat treatments during drying prior to storage. Drum-dried powders was dried at 105 °C for several seconds. Spray-dried powders was exposed for air at 72 °C for a few seconds in the drying chamber, whereas the freeze-dried powders were not exposed to any heat treatment at all. The chlorophyll degradation rate was different among the investigated powders during storage, with powders dried at high temperatures displayed limited absolute chlorophyll degradation. We suggest that the drum-dried powders did not had substantial amount of chlorophyll left after drying and the chlorophyllase were close to inactivated by the substantial heat treatment during drum-drying. Drum-dried powders were therefore omitted to one single degradation route of chlorophyll, the chemical degradation, which explains the limited chlorophyll degradation rate.

Freeze-dried powders were not exposed to heat during drying and displayed the highest chlorophyll degradation of the powders investigated. Chlorophyllase was intact and powder storage in 20 °C was a temperature close to optimal for the enzymatic activity [26], allowing a significant enzymatic reduction of chlorophyll. The degradation was more pronounced for higher water activities which is in accordance with Lajollo et al. [21]. Spray-dried powders were exposed for moderate heat during drying and displayed limited chlorophyll degradation at low relative humidity. At higher relative humidity the degradation rate was higher. We suggest that the moderate thermal treatment during spray-drying (72 °C for a few seconds) inactivated chlorophyllase to some extent. At low water activity, the enzymatic activity was inhibited but at higher water activities, the remaining chlorophyllase was accelerated by the free available water, causing significant chlorophyll degradation. Weemaes et al. reported that heat treatment at 50 °C accelerated the enzymatic activity in broccoli juice [27] but in the present study we found that dry thermal treatment at 72 °C for a few seconds instead resulted in limited chlorophyll degradation at low water activities and that the degradation was moisture dependent. We therefore suggest that the heat treatment during spray drying was harsh enough to partially inactivate chlorophyllase, rather than accelerate the enzymatic activity. Spray-dried powder displayed, at each relative humidity level investigated, the highest chlorophyll content of all powders. This may be due to the partial inactivation of chlorophyllase. We suggest that the chlorophyll in the spray-dried and freeze-dried powder was degraded via two routes: enzymatic catalyzed degradation known to be accelerated at higher relative humidity, and chemical acid-induced degradation. This can be the reason why spray-dried and freeze-dried powders were subjected to larger absolute chlorophyll degradation compared to drum-dried powders, although the chlorophyll content in spray-dried and freeze-dried powders was higher than the drum-dried powders at all relative humidity levels investigated. 

### 3.5. Effect of Storage on Thylakoids’ Interfacial Properties

Thylakoids’ emulsifying capacity of the all thylakoid powders was generally reduced after 8 months storage (Figure 5), and the loss was positively correlated to relative humidity. The effect was more pronounced for the spray-dried and freeze-dried powders compared to the drum-dried powders. Freeze-dried powders showed the most dramatic reduction in emulsifying capacity. Prior to incubation, the emulsifying capacity was 0.0116 m^2^/mg for freeze-dried powder and after 8 months storage in 61 RH% the emulsifying capacity was reduced to 0.00294 m^2^/mg, which corresponds to −75% loss.

The particle size of lipid droplets stabilized by thylakoid powder stored at 8 months in 61 RH% was increased compared to lipid droplets stabilized by thylakoid powders prior incubation (Figure 6). This indicates reduced emulsifying capacity of the membranes during storage in high relative humidity. The increase in emulsion droplet size was larger for spray-dried and freeze-dried powders compared with drum-dried powders. All powders except the spray-dried powder displayed one dominating peak. The spray-dried powder displayed two peaks, one corresponding to the same particle size as stabilized by thylakoids prior to incubation, and an additional peak with larger particle size. Emulsifying capacity of the thylakoids have previously been reported to be closely linked to chlorophyll content and lipase/co-lipase inhibiting effect [18]. Prolongation of lipolysis is in turn correlated to prolonged satiety in animal and human models demonstrated by Köhnke et al. 2009 [13,15]. Emulsifying capacity was therefore used as marker for the thylakoids’ functionality in the present study.

### 3.6. Chlorophyll Is Important for the Thylakoids’ Functional Properties

Chlorophyll has been suggested to play an important role in facilitating steric stabilization of hydrophobic alpha helices within the light harvesting complexes in the thylakoid membrane [33]. In a series of previous experiments, alpha helices have been identified as the surface-active parts [9], enable thylakoids to associate to the oil-water interface of lipid droplets, hindering lipase/co-lipase from reaching their substrate. When chlorophyll is degraded either by replacement of Mg^2+^ with 2 H^+^ or removal of the phytol chain, the polarity of the molecule is altered [29]. This could lead to reduced stability inside the thylakoid membrane, promoting association between hydrophobic domains [34]. Heat treatment of thylakoids induced chlorophyll degradation and reduced the lipase/co-lipase inhibiting capacity as well as the thylakoids’ emulsifying capacity as demonstrated in previous studies [18,23,24]. Aggregation of heat-treated thylakoids was observed in the previous studies, but it could not be concluded if the reason to the aggregation was chlorophyll degradation causing structural reorganization inside the membrane, or heat-induced aggregation of the thylakoid membranes. Degradation of chlorophyll may cause aggregation within the alpha helices known to play a decisive role in inhibition of lipase/co-lipase. Thylakoid membranes are also known to form inverted vesicles over 55 °C [35], with hydrophobic parts turned outside which facilitate aggregation due to minimization of free energy. The heat-induced aggregation can reduce the available thylakoid surface thereby reducing lipase/co-lipase inhibition.

In the present study, chlorophyll was degraded not by heat, but by enzymatic degradation with decreased emulsifying capacity as consequence. Thus, the present study indicates that chlorophyll is important for the internal structure of the thylakoid membranes. When chlorophyll was degraded, either by heat or enzymatic degradation, association between the hydrophobic alpha helices was suggested to be promoted, causing reduced surface activity of the thylakoid membranes, thus reduced emulsifying capacity.

Drum-dried thylakoids with low amount remaining chlorophyll were still able to stabilize the oil-water interface to a limited extent. It is therefore suggested that the thylakoids interfacial properties are determined by both a minor chlorophyll-independent factor and a major chlorophyll-dependent factor.

The results from the present study demonstrate that chlorophyll is important for the thylakoids function in terms of interfacial properties. Chlorophyll is known to be degraded by light, acid, heat, and by enzymes accelerated at higher relative humidity. To assure the interfacial properties of the thylakoids essential for lipase/co-lipase inhibition hence increased satiety, the chlorophyll content must be maintained to as great extent as possible during both processing and storage. For powders produced by drum drying, the chlorophyll was degraded by heat during drying and chlorophyllase was inactivated due to exposure to high temperatures, hence less degradation during storage. Thylakoid powders produced by the gentler drying methods freeze-drying and spray-drying had high chlorophyll levels after drying and were therefore sensitive for enzymatic chlorophyll degradation accelerated at higher relative humidity. The heat exposure during spray drying induced partial inactivation of chlorophyllase in the spray-dried powder. Therefore spray-dried powders had the highest levels of chlorophyll and highest emulsifying capacity at all relative humidity levels investigated, compared to the other powders.

## 4. Conclusions

The effect of 8 months storage in varying relative humidity on thylakoid powders dried by drum-drying, spray-drying or freeze-drying was evaluated. All powders were considered hygroscopic and absorbed moisture from the surrounding air. Addition of maltodextrin decreased the hygroscopicity of the drum-dried powder. The green color, chlorophyll content and emulsifying capacity of the powders were decreased during storage. The effect was moisture-dependent with a greater loss at higher relative humidity. The effect was more pronounced for the spray-dried and freeze-dried powders, which contained higher initial levels of green color and chlorophyll. These powders were therefore more susceptible to chlorophyll degradation during storage. The thylakoid powder should be stored dark in air-tight packages, preferably single dose sachets, to control the powders’ total exposure for moisture during the shelf life. The suggested packaging would ensure the thylakoid powders’ function as appetite-reducing ingredient after storage. The results provide knowledge regarding standardization and quality control of a functional food ingredient with appetite-reducing properties.

## Figures and Tables

**Figure 1 foods-09-00669-f001:**
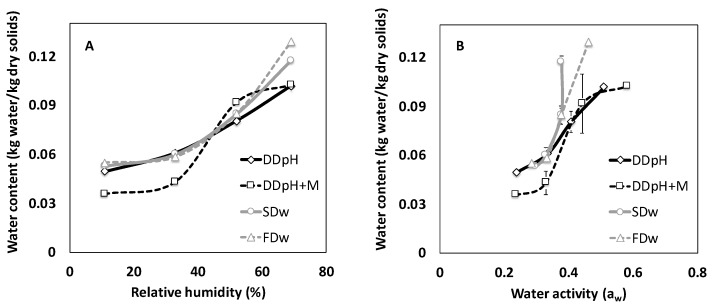
(**A**) Water content for thylakoid powders produced in different ways as a function of relative humidity in surrounding environment after 8 months of storage in controlled atmosphere with different relative humidity. (**B**) Water content for thylakoid powders produced in different ways as a function of water activity in the powders after 8 months of storage in controlled atmosphere with different relative humidity. DD_pH_ = drum-dried powder produced by the pH method, DD_pH+M_ = drum-dried powder produced by the pH method with addition of maltodextrin, SD_w_ = spray-dried powder produced by the water method, FD_w_ = freeze-dried powder produced by the water method.

**Figure 2 foods-09-00669-f002:**
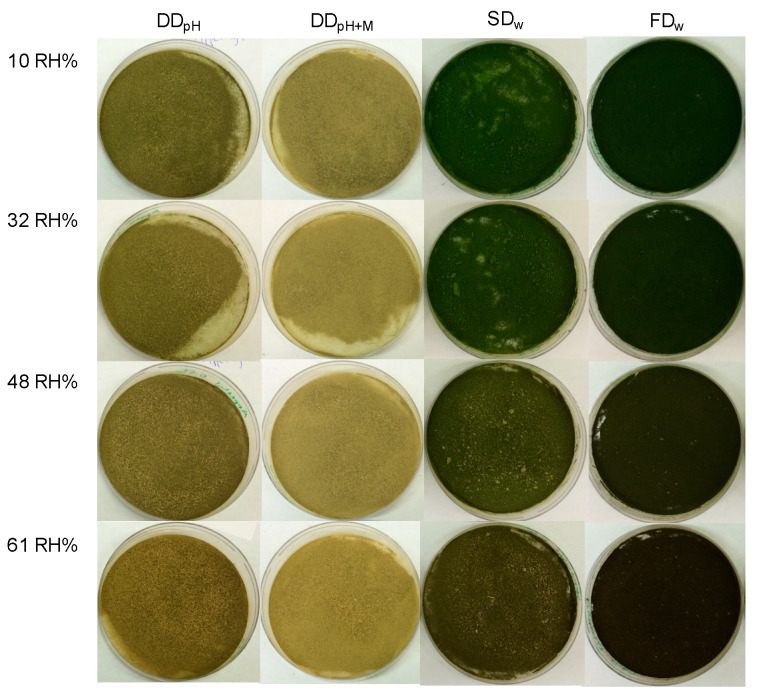
Photography of thylakoid powders produced in different ways after 8 months storage in controlled atmosphere with different relative humidity. DD_pH_ = drum-dried powder produced by the pH method, DD_pH+M_ = drum-dried powder produced by the pH method with addition of maltodextrin, SD_w_ = spray-dried powder produced by the water method, FD_w_ = freeze-dried powder produced by the water method.

**Figure 3 foods-09-00669-f003:**
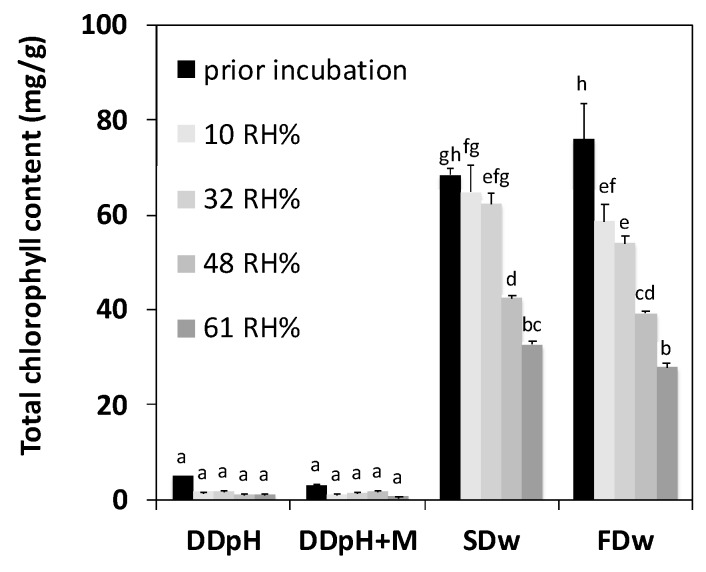
Total chlorophyll content of thylakoid powders produced in different ways after 8 months storage in controlled atmosphere with different relative humidity. DD_pH_ = drum-dried powder produced by the pH method, DD_pH+M_ = drum-dried powder produced by the pH method with addition of maltodextrin, SD_w_ = spray-dried powder produced by the water method, FD_w_ = freeze-dried powder produced by the water method. Different letters indicate significant difference (*p* < 0.05).

**Figure 4 foods-09-00669-f004:**
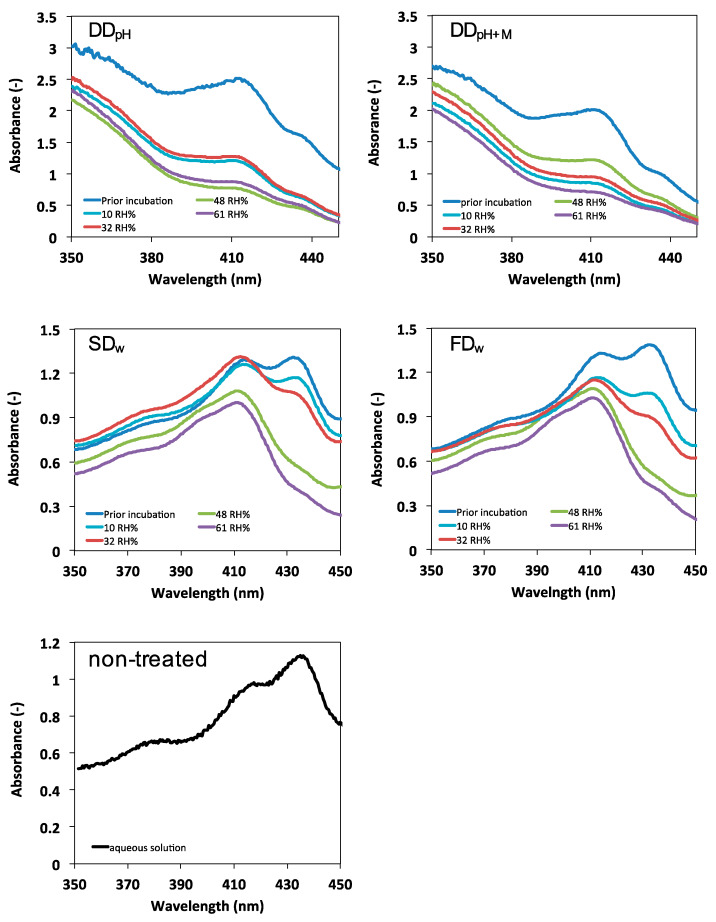
Absorption spectra for thylakoid powders produced in different ways after 8 months storage in controlled atmosphere with different relative humidity. Non-treated thylakoids in aqueous solution is provided for comparison. Due to limitations in the method, the absolute values cannot be compared. DD_pH_ = drum-dried powder produced by the pH method, DD_pH+M_ = drum-dried powder produced by the pH method with addition of maltodextrin, SD_w_ = spray-dried powder produced by the water method, FD_w_ = freeze-dried powder produced by the water method.

**Figure 5 foods-09-00669-f005:**
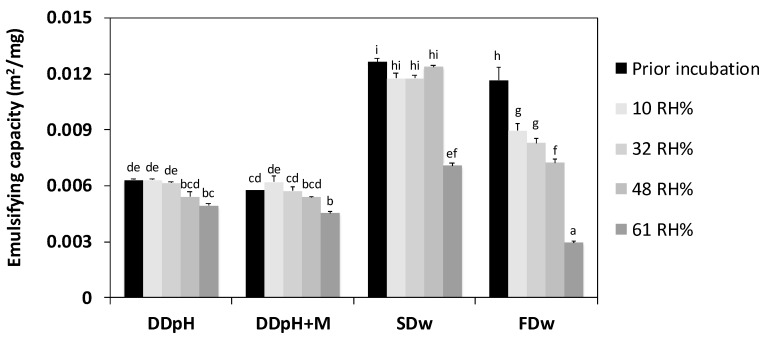
Emulsifying capacity of thylakoid powders produced in different ways after 8 months storage in controlled atmosphere with different relative humidity. DD_pH_ = drum-dried powder produced by the pH method, DD_pH+M_ = drum-dried powder produced by the pH method with addition of maltodextrin, SD_w_ = spray-dried powder produced by the water method, FD_w_ = freeze-dried powder produced by the water method. Different letters indicate significant difference (*p* < 0.05).

**Figure 6 foods-09-00669-f006:**
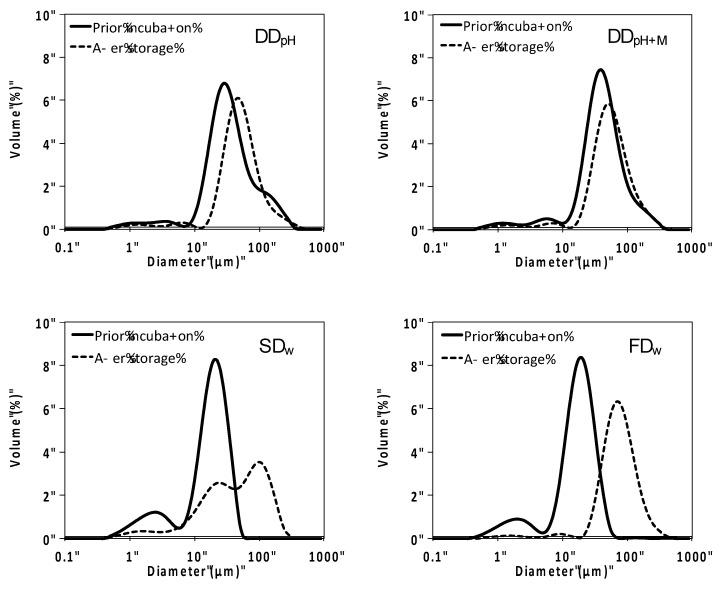
Particle size distribution of oil-in-water emulsions stabilized by thylakoid powder produced in different ways prior incubation and after 8 months storage in 61 RH%. DD_pH_ = drum-dried powder produced by the pH method, DD_pH+M_ = drum-dried powder produced by the pH method with addition of maltodextrin, SD_w_ = spray-dried powder produced by the water method, FD_w_ = freeze-dried powder produced by the water method.

**Table 1 foods-09-00669-t001:** Dry matter and color attributes for thylakoid powders dried in different ways after 8 months storage in varying relative humidity.

Label	Incubation in Relative Humidity	Dry Matter (%)	L	a	b
DD_pH_	Prior incubation	91.4 ± 0.5 ^aA^	23.0 ± 1.4 ^abAB^	−0.15 ± 0.0 ^aA^	20.0 ± 0.7 ^abA^
	RH 10%	95.3 ± 0.1 ^aA^	22.6 ± 1.0 ^abA^	−0.57 ± 0.1 ^aA^	15.2 ± 0.9 ^acA^
	RH 32%	94.3 ± 0.1 ^aA^	20.4 ± 0.2 ^aA^	−0.49 ± 0.0 ^aA^	20.6 ± 0.2 ^bA^
	RH 48%	92.6 ± 0.6 ^aA^	26.1 ± 0.1 ^abA^	0.48 ± 0.0 ^abA^	15.1 ± 0.3 ^aA^
	RH 61%	90.8 ± 0.1 ^aA^	31.5 ± 2.7 ^bA^	1.38 ± 0.2 ^bA^	14.1 ± 1.8 ^cAB^
DD_pH+M_	Prior incubation	96.9 ± 0.4 ^aB^	29.0 ± 2.4 ^aA^	−0.04 ± 0.0 ^abA^	17.0 ± 0.9 ^abcA^
	RH 10%	96.9 ± 0.4 ^aA^	24.3 ± 1.1 ^aA^	0.01 ± 0.1 ^abA^	13.3 ± 0.2 ^acA^
	RH 32%	95.9 ± 0.7 ^acA^	20.4 ± 0.2 ^aA^	−0.49 ± 0.0 ^aA^	20.6 ± 0.2 ^bcA^
	RH 48%	91.6 ± 1.5 ^bcA^	27.6 ± 1.4 ^aA^	0.60 ± 0.1 ^abA^	13.7 ± 0.8 ^aA^
	RH 61%	90.7 ± 0.2 ^bA^	29.6 ± 0.6 ^aA^	1.23 ± 0.1 ^bA^	16.2 ± 0.4 ^cA^
SD_w_	Prior incubation	95.7 ± 0.7 ^aB^	19.6 ± 1.7 ^abAB^	−9.25 ± 0.4 ^aB^	25.4 ± 1.6 ^aB^
	RH 10%	95.0 ± 0.1 ^abA^	12.8 ± 1.0 ^aB^	−9.02 ± 0.3 ^aB^	21.4 ± 1.5 ^abB^
	RH 32%	94.3 ± 0.4 ^abA^	15.4 ± 0.3 ^abA^	−8.41 ± 0.2 ^aB^	25.6 ± 0.5 ^aA^
	RH 48%	92.2 ± 0.5 ^bA^	22.9 ± 1.0 ^bcA^	−3.76 ± 0.1 ^bB^	20.2 ± 0.4 ^bB^
	RH 61%	89.5 ± 0.3 ^abAB^	31.6 ± 9.5 ^cA^	−0.58 ± 0.1 ^cB^	12.7 ± 4.0 ^cAB^
FD_w_	Prior incubation	97.7 ± 0.5 ^aB^	13.7 ± 0.9 ^aB^	−7.22 ± 01 ^aC^	16.0 ± 0.7 ^abA^
	RH 10%	94.8 ± 0.2 ^abA^	12.3 ± 1.8 ^aB^	−5.16 ± 1.0 ^bC^	12.6 ± 1.0 ^bcA^
	RH 32%	94.5 ± 0.2 ^abA^	n.a.	n.a.	n.a
	RH 48%	92.2 ± 0.3 ^abA^	12.9 ± 0.0 ^aB^	−1.98 ± 0.1 ^cC^	13.1 ± 1.0 ^bcA^
	RH 61%	88.6 ± 0.1 ^bAC^	13.4 ± 0.5 ^aB^	−0.07 ± 0.0 ^dAB^	10.9 ± 0.3 ^cB^

L, a and b are Hunters color coordinates. Labels: DD_pH_ = drum-dried powder produced by the pH method. DD_pH+M_ = drum-dried powder produced by the pH method with addition of maltodextrin. SD_w_ = spray-dried powder produced by the water method. FD_w_ = freeze-dried powder produced by the water method. Different uppercase letters indicate significant difference between incubation conditions within each powder formulation, *p* < 0.05. Different lowercase letters indicate significant difference within each incubation condition between powder formulations, *p* < 0.05.

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
