# Peer review of "Effects of Storage Conditions on Degradation of Chlorophyll and Emulsifying Capacity of Thylakoid Powders Produced by Different Drying Methods"

_foods, 2020, doi:10.3390/foods9050669_

Round 1

Reviewer 1 Report

Review

Manuscript ID: foods-797481

Title: Effects of storage conditions on degradation of chlorophyll and emulsifying capacity of thylakoid powders produced by different drying methods

The manuscript presents analysis of storage condition effect on the properties of thylakoid powders obtained by three drying methods: drum-drying, spray-drying and freeze-drying. Authors determined degradation of chlorophyll and emulsifying capacity after storage for 8 months in different controlled atmospheres. Presented research results have a chance to find application field in industrial practice.

Title is formulated clearly and correctly, writing style of whole article is clear and adapted to readers. Tables and graphics are easy to read and marked correctly. Bibliography is adequate to the topic, all included positions are cited in the text.

I recommend major revision.

Major Comments:

1. In reviewer’s opinion there are some key technological information missing about carried processes. Thus, it’s difficult to make an honest comparison of applied methods of thylakoid powder production and also to draw the right conclusions.

  • what was solid content in the solutions before drying process in each of methods ?
  • what was the exact way of obtaining the powder in drum and freeze drying methods? obtained dried material was processed by milling or not? Was there any screening process using sieves with specified mesh size?

  • there was no analysis of powder particle size for different production methods, which must be considered as missed opportunity in accordance to fact that Malvern Mastersizer 2000 was available to use.
  • why maltodextrin was used only in case of drum drying? What was the idea behind it?
  • what was the exact idea of applied research schedule and drying process parameters? 
  • why the 8 months storage period was applied? What was the idea?

Parameters and technological issues mentioned above are very important regarding obtained food and functional products in form of powders. Therefore not giving them may cause doubts about conclusions – whether they are correct or too general. Comparison of different processes must be based on specific logical assumptions that are clearly presented to reader. In some sort of conditions a comparison of different processes can be pointless.

2.  Why the authors applied different methods of thylakoid membranes isolation for various drying methods? Was the same raw material used for different methods of thylakoid membranes isolation? (fresh, frozen or hot air-dried spinach leaves ?) What was the idea for method and raw material selection?

3.  Section 3. Results and discussion. 

There was only one place in whole Results Discussion section where ‘%’ value was used to show degradation and loss level during 8 months storage period. The analysis of relative values given in percent can lead to different conclusions.

i.e.: Figure 3 – if statistical analysis was applied only within the results for DDpH sample, it would most likely appear that this specific sample showed statistically significant changes and the biggest relative loss level of total chlorophyll content - not SDw or FDw samples. That’s why it is so important to present all assumptions, parameters and research ideas so that readers could follow this.

4.  Section 3.4. Heat treatment during drying affects chlorophyll loss during storage.

The entire discussion in this section is vague. The authors compare different processes but it is not really known what temperature the dry powder particles have reached. Was the powder temperature measured immediately after each process? Each process has a different heat transfer mechanism. For example, it is not possible to compare the outlet air temperature in spray drying to the temperature of juice heating used in Weemaes' study [27] (line 338-340).

Minor comments:

Line 69: „ … by spray-dried (72°C for a few seconds)” – given temperature is not accurately defined. It is most likely the temperature of outlet air, however more common way is to give inlet air temperature. The most favorable solution is to give both – 120°C /72°C;

Line 98: „… of polyester mesh filter” – mesh size wasn’t defined;

Line 108: “… centrifuged at 5000 g, …” – in 156-157 lines it is written: “centrifugation (…) at 13400 x g” – please unify (with use of ‘x’ or without it);

Line 162-163: whether the presented equations were determined by the authors or based on literature ?

Line 210-211: „Drum-dried powder with addition of maltodextrin (20% w/w) …”- an information that maltodextrin was added in quantity of 1:3 w/w dry base was given in 2.2 section (Powder production) – these are inconsistent values, please check and correct;

Line 217: „Water content varied from 0.0315 kg water/kg solids (10 RH%) …” – in accordance to table 1 it is not a value for incubation in RH 10% but for “prior incubation” sample;

Line 218: “… 0.0233 kg water/ kg solids (10 RH%) …” – the same as above;

Table 1:

Please check dry matter and water content values for DDpH sample – they seem to be incorrect;

Figure 2:

Why the authors didn’t included photos of „prior incubation” sample to the drawing for comparison?

Line 289-290: “Simultaneously the peak corresponding to the degradation products pheophorbide a and pheophytin a (both displaying absorption maximum at 409-415 nm [19]) was increased (Figure 4)” - is this sentence really true ? Is it the peak increase for SDw and FDw after storage or compared to drum drying (DDpH) ?.

Figure 6:

Please check and correct chart legends.

Author Response

Dear Reviewer #1!

Thanks for the valuable comments. We have done our best to make the suggested changes and hope you find them satisfactory in the revised manuscript. Please find our responses to the comments below. In the revised manuscript new and significantly modified sections are indicated in green.

Title: Effects of storage conditions on degradation of chlorophyll and emulsifying capacity of thylakoid powders produced by different drying methods

The manuscript presents analysis of storage condition effect on the properties of thylakoid powders obtained by three drying methods: drum-drying, spray-drying and freeze-drying. Authors determined degradation of chlorophyll and emulsifying capacity after storage for 8 months in different controlled atmospheres. Presented research results have a chance to find application field in industrial practice.

Title is formulated clearly and correctly, writing style of whole article is clear and adapted to readers. Tables and graphics are easy to read and marked correctly. Bibliography is adequate to the topic, all included positions are cited in the text.

I recommend major revision.

Major Comments:

In reviewer’s opinion there are some key technological information missing about carried processes. Thus, it’s difficult to make an honest comparison of applied methods of thylakoid powder production and also to draw the right conclusions.

Comment #1: what was solid content in the solutions before drying process in each of methods?

Response: The soild content before the drying process was 0.011 kg dry solids/kg water and the details have been added to the Material and method section (line 97-98).

Comment #2: what was the exact way of obtaining the powder in drum and freeze drying methods? obtained dried material was processed by milling or not? Was there any screening process using sieves with specified mesh size?

Response: Thanks for the comment. The drum dried powders were provided from a company, Green Leaf Medicals, and the production conditions cannot be shared in details although the conditions has been reported broadly in the manuscript. There must have been a milling step included but details are not known to the authors. The method section has been rewritten in order to clarify that it is two commercial samples and two samples prepared in house in our lab (lines 92-123) However, the freeze dried samples were grinded in an electrical coffee miller. Details regarding the grinding step for freeze dried powder is now included in the Material and Method section (lines 106-107). We performed an attempt to sieve the samples, but it took a non-zero amount of time to get the powder through the sieves and DDpH, DDpH+M and the freeze-dried powder absorbed a significant amount of water. Since we wanted the powder to have as equal water content as possible before onset of the experiment in the different relative humidity, we decided to go for non-sieved samples instead.  

Comment #3: there was no analysis of powder particle size for different production methods, which must be considered as missed opportunity in accordance to fact that Malvern Mastersizer 2000 was available to use.

Response: We agree and will include particle size in our studies in the future.

Comment #4: why maltodextrin was used only in case of drum drying? What was the idea behind it?

Response: The drum dried samples were provided from a company and they have two different products: one with added maltodextrin and one without. We wanted to include both products in the present study. Details about the commercial samples have been added (lines 112-123) 

Comment #5: what was the exact idea of applied research schedule and drying process parameters? 

Response: The idea was to compare two industrially produced thylakoid samples with samples prepared in our lab.

Comment #6: why the 8 months storage period was applied? What was the idea?

Response: We wanted to investigate a longer period of time not only a few weeks or months. 

Parameters and technological issues mentioned above are very important regarding obtained food and functional products in form of powders. Therefore not giving them may cause doubts about conclusions – whether they are correct or too general. Comparison of different processes must be based on specific logical assumptions that are clearly presented to reader. In some sort of conditions a comparison of different processes can be pointless.

Comment #7: Why the authors applied different methods of thylakoid membranes isolation for various drying methods? Was the same raw material used for different methods of thylakoid membranes isolation? (fresh, frozen or hot air-dried spinach leaves ?) What was the idea for method and raw material selection?

Response: We compared two commercial samples with the method we have developed in our lab. The commercial samples were prepared with the pH method and drum-dried and the raw material for these products were air-dried spinach. The samples we prepared in the lab was prepared with the water method with frozen spinach leaves as raw material and we wanted to investigate two different drying methods: spray drying and freeze-drying and how that is affecting the quality during storage. The aim has been rephrased (lines 85-89) and also the Material and method section has been rewritten in order to make it clearer to the reader (lines 92-123).

Section 3. Results and discussion. 

Comment #8: There was only one place in whole Results Discussion section where ‘%’ value was used to show degradation and loss level during 8 months storage period. The analysis of relative values given in percent can lead to different conclusions.

Response: Thanks for the comment, the result and discussion section has been improved to also reflect the statistical analysis (lines 207-275, 297-306).

Comment #9: i.e.: Figure 3 – if statistical analysis was applied only within the results for DDpH sample, it would most likely appear that this specific sample showed statistically significant changes and the biggest relative loss level of total chlorophyll content - not SDw or FDw samples. That’s why it is so important to present all assumptions, parameters and research ideas so that readers could follow this.

Response: Thanks for the comment. The statistical method we used (General linear model) is comparing every sample against each other. In other terms – there are no statistical differences within the group DDpH or within the group DDpH+M.

Section 3.4. Heat treatment during drying affects chlorophyll loss during storage.

Comment #10: The entire discussion in this section is vague. The authors compare different processes but it is not really known what temperature the dry powder particles have reached. Was the powder temperature measured immediately after each process? Each process has a different heat transfer mechanism. For example, it is not possible to compare the outlet air temperature in spray drying to the temperature of juice heating used in Weemaes' study [27] (line 338-340).

Response: We totally agree that one cannot directly compare the wet and dry temperature and has reformulated the section to make it more clear (lines 367-372). We did not measure the powder temperature per se, but we logged the temperature of the outgoing air in the spray drier and since evaporation is a cooling process and the powder particle in spray drying is not exposed to temperatures above the outlet temperature we decided to report this temperature in the manuscript. The drum-dried powders were commercial samples and we do not have access to further details more than that the powders are drum-dried with a drum surface temperature around 105°C. 

Minor comments:

Comment #11: Line 69: „ … by spray-dried (72°C for a few seconds)” – given temperature is not accurately defined. It is most likely the temperature of outlet air, however more common way is to give inlet air temperature. The most favorable solution is to give both – 120°C /72°C;

Response: Thanks for the comments, we have added the inlet temperature for the study we are referring to in the introduction (lines 69-70).

Comment #12: Line 98: „… of polyester mesh filter” – mesh size wasn’t defined;

Response: The drum-dried powders were commercial samples and we do not have access to details regarding the filtration. The method section has been rewritten to reflect this.

Comment #13: Line 108: “… centrifuged at 5000 g, …” – in 156-157 lines it is written: “centrifugation (…) at 13400 x g” – please unify (with use of ‘x’ or without it);

Response: Thanks for the comments, we have deleted “x” to be consistent.

Comment #14: Line 162-163: whether the presented equations were determined by the authors or based on literature ?

Response: The equations are determined by Porra et al and the reference has been added (line 158)

Comment #15: Line 210-211: „Drum-dried powder with addition of maltodextrin (20% w/w) …”- an information that maltodextrin was added in quantity of 1:3 w/w dry base was given in 2.2 section (Powder production) – these are inconsistent values, please check and correct;

Response: The addition of maltodextrin in the commercial sample was 1:3 w/w dry base and we have changed the incorrect concentration to report this ratio (line 216).

Comment #16: Line 217: „Water content varied from 0.0315 kg water/kg solids (10 RH%) …” – in accordance to table 1 it is not a value for incubation in RH 10% but for “prior incubation” sample;

Response: Thanks for the valuable comment. The correct values (0.0528 kg water/kg solids for spray-dried powders and 0.0528 kg water/kg solids for freeze-dried powders i incubated in 10 RH%) are now inserted at lines 225-226.

Comment #17: Line 218: “… 0.0233 kg water/ kg solids (10 RH%) …” – the same as above;

Response: The values have been corrected (lines 225-226).

Comment #18: Table 1: Please check dry matter and water content values for DDpH sample – they seem to be incorrect;

Response: Dry matter values and water content values have been corrected. According to one of the other reviewers, water content and water activity is removed from Table 1 to not repeat data.The data in Figure 1 is correct.   

Comment #19:  Figure 2: Why the authors didn’t included photos of „prior incubation” sample to the drawing for comparison?

Response: Thanks for the comment, unfortunately no photography’s from time 0 is available. We will include photos of the starting material in our future studies. Honestly, we did not expect the color change to be so dramatic as it turned out and if we have knew we would definitely been taking photos before. However, we measured the L, a and b values both prior incubation and after incubation which is reported in Table 1. 

Comment #20: Line 289-290: “Simultaneously the peak corresponding to the degradation products pheophorbide a and pheophytin a (both displaying absorption maximum at 409-415 nm [19]) was increased (Figure 4)” - is this sentence really true ? Is it the peak increase for SDw and FDw after storage or compared to drum drying (DDpH) ?.

Response: Thanks for the valuable comment, you are totally correct, and the paragraph has been reformulated to avoid misunderstandings (lines 316-321).

Comment #21: Figure 6: Please check and correct chart legends.

Response: We have doublechecked the chart legends.

Reviewer 2 Report

The results are original, but some of the results were not properly elaborated and analyzed.

The article requires some explanations and complements, and then checking the discussion of the results:

line 139-142: please provide the determination conditions (temperature?)

Fig.1 Explain which values of water activity (x axis) were used in drawing up this Figure? The same aw values as in Table 1? Why not RH values?

In my opinion they are not "Moisture sorption isotherms".

A moisture sorption isotherm for a food product is the relationship between water activity (aw) and moisture content at a given temperature. This relationship is complex and unique for each product due to different interactions between the water and the solid components at different moisture contents. Traditional isotherm methods depend on establishing the equilibration of samples to known water activity values. Six to nine different humidity levels are needed and vapor equilibration must be achieved.

However, based on the data in Table 1, we do not know whether equilibrium has been reached (compare RH values with aw values).

In addition, the data in Table 1 show that the DDpH sample was wet, and consequently, desorption and not adsorption took place in this sample.

Table 1:

- delete the third column - water content data has already been used in Figure 1;

- some L a * b * values are illogical (e.g. for FDw at RH32%), check them;

- what about HUE and chroma?

Author Response

Dear Reviewer #2! 

Thanks for the valuable comments. We have done our best to make the suggested changes and hope you find them satisfactory in the revised manuscript. Please find our responses to the comments below. In the revised manuscript new and significantly modified sections are indicated in green. We feel that the comments have improved the manuscript significantly.

The results are original, but some of the results were not properly elaborated and analyzed.

The article requires some explanations and complements, and then checking the discussion of the results:

Comment #1: line 139-142: please provide the determination conditions (temperature?)

Response: The experiments was conducted in a room with controlled temperature and the temperature was 20°C during the entire experiment (line 130). 

Comment #2: Fig.1 Explain which values of water activity (x axis) were used in drawing up this Figure? The same aw values as in Table 1? Why not RH values?

Response: Thanks for the comment, the values in Fig 1 is the same as the aw values in Table 1 even though there were a few typing mistakes but the data in the figure is correct. According to response from one of the other reviewers we have deleted water activity and moisture content from Table 1 to not repeat data. We also added a figure with relative humidity on the x-axes to give the full picture. We want to show the water content as a function of water activity as well, since that is what was measured in the samples. When the two figures are compared it is clear that DDpH+M (the powder with maltodextrin) has reach equilibrium after 8 months but that the other powders have not. This is now included in the results and discussion section (lines 215-216)

Comment #3:  In my opinion they are not "Moisture sorption isotherms". A moisture sorption isotherm for a food product is the relationship between water activity (aw) and moisture content at a given temperature. This relationship is complex and unique for each product due to different interactions between the water and the solid components at different moisture contents. Traditional isotherm methods depend on establishing the equilibration of samples to known water activity values. Six to nine different humidity levels are needed and vapor equilibration must be achieved.

However, based on the data in Table 1, we do not know whether equilibrium has been reached (compare RH values with aw values).

In addition, the data in Table 1 show that the DDpH sample was wet, and consequently, desorption and not adsorption took place in this sample.

Response: Thanks for the comment, the purpose of the study was to investigate how powder characteristics and emulsifying capacity (which is linked to enzymatic inhibition of lipase-co-lipase) changed over time in different relative humidities in order to simulate what would happen if an thylakoid appetite-regulating substance was stored in dry or humid atmosphere. We agree that three of the samples have not reached equilibrium (DDpH, SD and FD) during 8 months of storage but by this approach we could investigate if the samples were stable or if the water content was changing over time and the results can be used for recommendations for storage conditions. The results section has been updated to also include a comment about this (line 215-216). We have also changed the figure legend: (a) Water content for thylakoid powders produced in different ways as a function of relative humidity in surrounding environment after 8 months of storage in controlled atmosphere with different relative humidity. (b) Water content for thylakoid powders produced in different ways as a function of water activity in the powders after 8 months of storage in controlled atmosphere with different relative humidity. DDpH and DDpH+M are commercial samples from a company and DDpH had indeed much higher water content from the beginning. The Material and method section have been rewritten to make it clearer to the reader (lines 112-123). Regarding the DDpH sample, the result section has been rewritten and a comment on desorption is now included (line 208-214) 

Comment #4: - delete the third column - water content data has already been used in Figure 1;

Response: We have deleted the water content data to not repeat data.

Comment #5: - some L a * b * values are illogical (e.g. for FDw at RH32%), check them;

Response: Thanks for the valuable comment. We have double checked the lab journal and all the replicates noted for FWw at 32 RH% are around 3 for L-value. We agree that something is very wrong here with this specific sample and have changes to “not available” in the Table. We have double checked the L, a and b values and the correct values are now presented in Table 1.

Comment #6: what about HUE and chroma?

Response: We chose to report the values in L, a and b values since it is an established method to present green colour. By this approach we could follow the degradation of chlorophyll in two different ways.

Reviewer 3 Report

The research is very interesting; it has an important scientific and practical dimension.

Purposefulness of the conducted study was demonstrated and justified. The Authors noted that thylakoid membranes isolated from spinach can be used as an appetite-reducing ingredient promoting weight loss due to increased satiety between meals. The choice of technology for their preservation can be of great importance in the production of functional food. Therefore, they conducted the study to develop stable thylakoid products as desired properties and functionality after processing by drying method to powder form, and also during storage. The effect of storage in 4 varying relative humidity on thylakoid powders obtained by drum-drying, spray-drying and freeze-drying was evaluated. Powder characteristics and functionality, i.e. dry matter, water activity, chlorophyll content, colour and emulsifying capacity were evaluated after 8 months storage.

In subsequent studies, it is worth analysing other properties, typical for powders, and concerning their stability, e.g. tendency to caking, flowability, bulk density..

The manuscript prepared properly, well written and organized. I have no comments to the title, introduction, materials and methods, experimental procedures, references (although only 2 items are from the last 5 years; from 2015 and 2018). It is important to underline the extensive discussion of the results. Statistical analysis is shown in the discussion of the results, but it is missing in the table. The conclusions are supported by evidence. References contain 35 sources relevant to the subject. The language is correct.

Other comments:

Page 3, line 124: Is the pressure correct? “the vacuum pressure of the dryer was 0.02 mbar”

In table 1 it is worth adding the details of the statistical analysis (homogeneous groups).

Author Response

Dear Reviewer #3!

Thanks for the valuable comments. We have done our best to make the suggested changes and hope you find them satisfactory in the revised manuscript. Please find our responses to the comments below. In the revised manuscript new and significantly modified sections are indicated in green. We feel that the comments have improved the manuscript significantly.

The research is very interesting; it has an important scientific and practical dimension.

Purposefulness of the conducted study was demonstrated and justified. The Authors noted that thylakoid membranes isolated from spinach can be used as an appetite-reducing ingredient promoting weight loss due to increased satiety between meals. The choice of technology for their preservation can be of great importance in the production of functional food. Therefore, they conducted the study to develop stable thylakoid products as desired properties and functionality after processing by drying method to powder form, and also during storage. The effect of storage in 4 varying relative humidity on thylakoid powders obtained by drum-drying, spray-drying and freeze-drying was evaluated. Powder characteristics and functionality, i.e. dry matter, water activity, chlorophyll content, colour and emulsifying capacity were evaluated after 8 months storage.

In subsequent studies, it is worth analysing other properties, typical for powders, and concerning their stability, e.g. tendency to caking, flowability, bulk density..

The manuscript prepared properly, well written and organized. I have no comments to the title, introduction, materials and methods, experimental procedures, references (although only 2 items are from the last 5 years; from 2015 and 2018). It is important to underline the extensive discussion of the results. Statistical analysis is shown in the discussion of the results, but it is missing in the table. The conclusions are supported by evidence. References contain 35 sources relevant to the subject. The language is correct.

Other comments: 

Comment #1: Page 3, line 124: Is the pressure correct? “the vacuum pressure of the dryer was 0.02 mbar”

Response: Thanks for the comment, it should be ”the pressure in the freeze dryer was 0.02 mbar” and we have corrected it in the manuscript (line 105)

Comment #2: In table 1 it is worth adding the details of the statistical analysis (homogeneous groups).

Response: The data in the table are now statistically evaluated, and statistical results are included in the result and discussion section.

Reviewer 4 Report

Materials and methods

pag 2 line 96: please write some detalis about the hot-air-drying of spinach raw materials.

pag 2 98: "...were filtered through four layers of polyester mesh filter..", please write the μm.

pag 2 line 104: are the spinach leaves dried?

Results and discussion

All discussion is poor and in all paragraphs the comments about the statistical analysis are missing. 

In the paragraph 3.1 and in the table 1 the statistical discussion in terms of significaty is missing

Figure 2. The Photography of thylakoid powders produced in different ways at time 0 of storage are necessary 

the authors have to report the colour values during storage, the photography is not enough.

Author Response

Dear Reviewer #4!

Thanks for the valuable comments. We have done our best to make the suggested changes and hope you find them satisfactory in the revised manuscript. Please find our responses to the comments below. In the revised manuscript new and significantly modified sections are indicated in green. We feel that the comments have improved the manuscript significantly.

Materials and methods

Comment #1: pag 2 line 96: please write some detalis about the hot-air-drying of spinach raw materials.

Response: Thanks for the comment. The drum dried powders were provided from a company, Green Leaf Medicals, and the production conditions cannot be shared in details although the conditions has been reported broadly in the manuscript. The Material and method section have been reformulated to make it more clear for the reader (lines 92-123).

Comment #2: pag 2 98: "...were filtered through four layers of polyester mesh filter..", please write the μm.

Response: The paragraph has been rewritten to reflect that it is a commercial sample, and details about mesh is not available for the authors (lines 112-123).

Comment #3: pag 2 line 104: are the spinach leaves dried?

Response: Thanks for the comment, the samples called SD and FD was prepared in our lab and frozen spinach leaves were used (not dried). We have clarified that the spinach was frozen (line 92)

Results and discussion

Comment #4: All discussion is poor and in all paragraphs the comments about the statistical analysis are missing. 

Response: The discussion has been rewritten and results from the statistical analysis has been added (lines 207-237, 257-275, 297-306)

Comment #5: In the paragraph 3.1 and in the table 1 the statistical discussion in terms of significaty is missing

Response: Statistical discussion has been added (lines 207-237, 257-275, 297-306)

Comment #6: Figure 2. The Photography of thylakoid powders produced in different ways at time 0 of storage are necessary 

Response: Thanks for the comment, unfortunately no photography’s from time 0 is available. We will include photos of the starting material in our future studies. Honestly, we did not expect the color change to be so dramatic as it turned out and if we have knewn we would definitely been taking photos before. However, we measured the L, a and b values both prior incubation and after incubation which is reported in Table 1. 

Comment #7: the authors have to report the colour values during storage, the photography is not enough.

Response: L, a and b values are provided in table 1 to give color coordinates.

Round 2

Reviewer 1 Report

The authors responded to all my comments in the previous review and made relevant corrections to the manuscript. I recommend the article in its current form for publication in Foods.

Reviewer 4 Report

The authors improved completely the manuscript. Now it is clear and well written. For this reason It can be accepted in the present form.